# Clinical and Prognostic Significance of TZAP Expression in Cervical Cancer

**DOI:** 10.3390/medicina56050207

**Published:** 2020-04-25

**Authors:** Won-Jin Park, Jae-Hee Park, Ho-Yong Shin, Jae-Ho Lee

**Affiliations:** 1Department of Anatomy, School of Medicine Keimyung University, Dalgubeoldaero, Dalseo-Gu, Daegu 2800, Korea; dnjswls824@naver.com (W.-J.P.); cpr8282@dsmc.or.kr (J.-H.P.); 2Medical Course, School of Medicine Keimyung University, Dalgubeoldaero, Dalseo-Gu, Daegu 2800, Korea; purme2000@naver.com

**Keywords:** TZAP, telomere, cervical cancer, ZBTB48

## Abstract

*Background and Objectives*: Telomeric zinc finger-associated protein (TZAP) is a telomere-associated factor that was previously called ZBTB48. This protein binds preferentially to long telomeres, competing with telomeric repeat factors 1 and 2. Genetic changes in TZAP may be associated with cancer pathogenesis; however, this relationship has not yet been elucidated for any type of cancer. In this study, we aimed to examine the clinicopathologic and prognostic value of TZAP expression in cervical cancer (CC). *Materials and Methods*: The data were extracted from The Cancer Genome Atlas cohorts by OncoLnc (21 cancer types, 7700 cancers). The prognostic value of TZAP for different stages of 264 CCs was examined using survival analysis. *Results*: The TZAP expression did not differ significantly between CC and normal matched tissues. Age, cancer stage, and viral infection were not associated with TZAP expression. Survival analysis revealed a shorter overall survival in CC patients with a lower TZAP expression (χ^2^ = 3.62, *p* = 0.057). The prognostic value of TZAP expression was greater in patients with N1 stage CC (χ^2^ = 5.64, *p* = 0.018). *Conclusion*: TZAP expression is a possible prognostic marker for CC, especially stage N1 CC.

## 1. Introduction

Telomeres, as nucleoprotein complexes, are composed of TTAGGG repeat sequences [1]. Typically, telomeres in somatic cells are shortened by about 0–200 base pairs for every cell division and finally reach a critical length, inducing replicative senescence and apoptosis [2,3]. Therefore, the maintenance of the optimal range of telomere length (TL) is essential for cell survival [1,2,3].

Telomere shortening is counteracted by the reverse transcriptase telomerase in stem cells and most cancers, and an alternative lengthening mechanism is also worked in other cancers [4,5]. Due to this process for telomere elongation, overly long telomeres are identified and are then cut to a normal length by rapid trimming [6]. A recent article introduced the zinc finger protein ZBTB48 (renamed as telomeric zinc finger-associated protein, TZAP), suggesting that its overexpression causes progressive telomere shortening [7]. While it indicated that genetic alterations of TZAP may influence cancer pathogenesis, smaller studies examining this have been performed in some cancers [8,9]. These studies suggested that insufficient TZAP expression or its disorder caused telomere elongation in immortal cancers cells, which predicted poorer prognosis.

Cervical cancer (CC) constitutes a significant public health problem and is the main cause of death due to malignant tumors in women [10,11]. Many factors contributing to CC development have been identified, including human papillomavirus infection, smoking, and the use of oral contraceptives [12,13]. However, genetic studies in CC are rare and the clinical significance of genetic changes in CC remains unknown. 

This study aimed to examine TZAP expression in cohorts of patients with different types of cancer, with a focus on well-defined primary CC, and to investigate the clinicopathologic and prognostic value of TZAP expression in this type of cancer. The study for clinicopathological characteristics of TZAP may present the clue for its prognostic value. 

## 2. Methods

Primary data from The Cancer Genome Atlas (TCGA) data portal were downloaded in March 2020 [11]. The prognostic value of TZAP was compared in 21 cancer types using mRNA expression data and clinical information from OncoLnc [14], after dividing cancer patients into two groups according to the median value of TZAP expression (high vs. low expression). It provided the rank of the *p* value for TZAP expression in each cancer type. The cancer type showing the most promising results, CC, was then selected, and a detailed analysis of it was performed.

Overall survival (OS) was defined as the duration from the date of surgery to the date of the last follow-up visit or the date of death due to any cause, whereas disease-free survival (DFS) was defined as the duration from surgery to any type of recurrence. 

All statistical analysis was performed with Statistical Package for the Social Sciences (SPSS), version 25.0 for Windows (IBM, Armonk, NY, USA). Chi-square and Mann–Whitney U-tests were used to analyze for categorical variables and continuous variables, respectively. Survival curves, constructed using univariate Kaplan–Meier estimators, were compared using the log-rank test. A two-tailed *p* value of <0.05 was considered to signify statistical significance.

## 3. Results

### 3.1. TZAP Expression in Different Cancer Types

The prognostic value of TZAP in different cancer types is presented in Figure 1. TZAP expression showed a good prognostic value in bladder, breast, colon, and pancreatic cancers. While it did not show a high statistical value for CC genes (*p* = 0.0529), its P-value was particularly high for this cancer type (No. 158 of all registered cancers). In comparing the TZAP expression between the bottom and top thirds and between the bottom and top quartiles, CC showed a significant value in both the third and quartile sorting (*p* < 0.001). Therefore, the clinical and prognostic value of TZAP expression in CC were further studied.

### 3.2. TZAP Expression in Cervical Cancer

The TZAP expression between non-cancerous and CC tissues was similar (8.80 ± 0.19 vs. 8.86 ± 0.52; *p* = 0.782, Figure 2). The clinicopathological characteristics of TZAP expression in CC are summarized in Table 1. TZAP expression was higher in CC patients with a lower T stage, although the difference was not statistically significant (*p* = 0.251). Other characteristics were not associated with TZAP expression in CC.

The median follow-up period in the cohort examined in the survival analysis was 36.2 months (range: 1–213.6 months). Univariate survival analysis revealed a shorter OS in CC patients with a lower TZAP expression (χ^2^ = 3.62, *p* = 0.057; Figure 3A). When patients were stratified according to the N stage, the prognostic value of TZAP expression appeared to be greater. In N1 stage patients, a lower TZAP expression was a significant predictor for a poorer OS (χ^2^ = 5.64, *p* = 0.018; Figure 3B). However, TZAP did not show a good predictive power for DFS in CC (χ^2^ = 0.099, *p* = 0.753).

## 4. Discussion

To our knowledge, this study is the first to examine TZAP expression in CC. TZAP expression may be important for telomere maintenance in cancer [7]. When TZAP expression is insufficient in cancer cells, overly long telomeres can easily change into immortal cells, predicting cancer progression [15,16]. Therefore, TZAP could mediate telomere trimming in cancers with abnormal telomere lengthening and it may be associated with cancer biology. However, TZAP expression in cancers has not been examined, and this hypothesis remains untested. 

TCGA big data suggested that TZAP may have an important role in determining CC prognosis, as well as in pancreatic and colorectal cancers [17]. However, we focused on CC to identify the clinical and prognostic value of TZAP expression. Contrary to our expectations, we found that TZAP expression did not have any clinicopathological or prognostic significance. However, in N1 stage patients a lower TZAP expression was associated with a poorer survival, which is consistent with the previous hypothesis that an insufficient TZAP expression could generate long telomeres, thus inducing cancer progression via immortal cells [7,15,16]. Clinically, N staging did not have significant value, however, this finding contributes to the role of telomeres in the pathogenesis of CC. The exact mechanism of this remains unclear and should be examined further in different cancer types. 

Our previous study showed that TZAP mutations were present in breast cancer and were associated with the N stage and a longer telomere length [9]. Interestingly, this tendency was found in breast cancer and CC as representative women’s cancers, and it should be studied in other cancers. These mutations also resulted in a poorer survival, thereby supporting our results and hypothesis. We further demonstrated that TZAP expression was positively correlated with telomerase reverse transcriptase expression in various cancer types [8]. Moreover, TZAP appears to compete with telomeric repeat factors 1 and 2 in binding to telomeres. This data suggested that TZAP may closely play with other telomere regulation genes. It should be confirmed by a translational study with patient samples. 

## 5. Conclusions

Taken together, these results suggest that the regulation of telomere length, which is mediated by these proteins, is an extremely complicated process. Therefore, further studies should aim to identify the molecular mechanisms underlying the telomere-related crosstalk between TZAP and other cellular processes. 

## Figures and Tables

**Figure 1 medicina-56-00207-f001:**
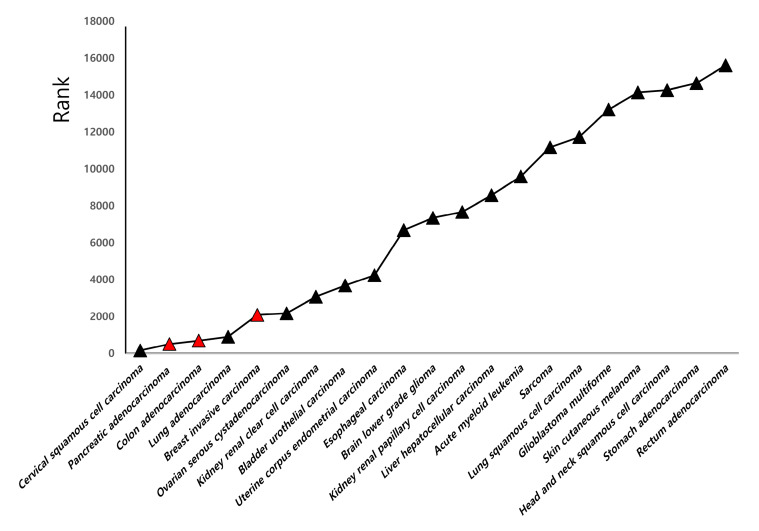
The rank of prognostic value of TZAP expression in different cancers. Red triangle, *p* < 0.05.

**Figure 2 medicina-56-00207-f002:**
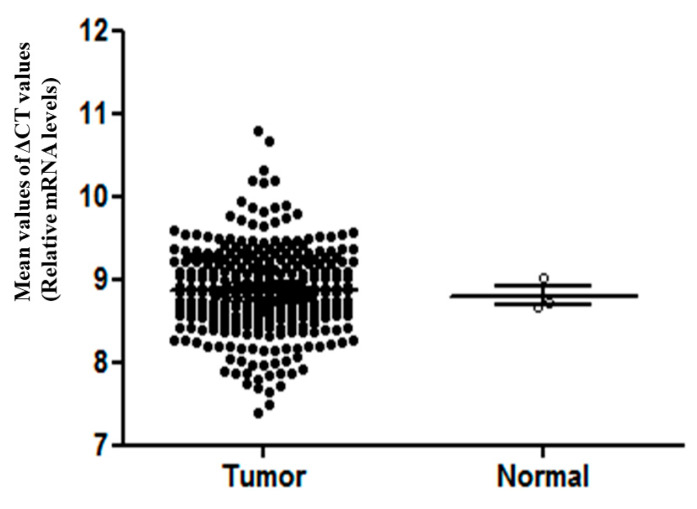
TZAP expression in cervical cancer and non-cancerous lesions. Mann–Whitney U-test.

**Figure 3 medicina-56-00207-f003:**
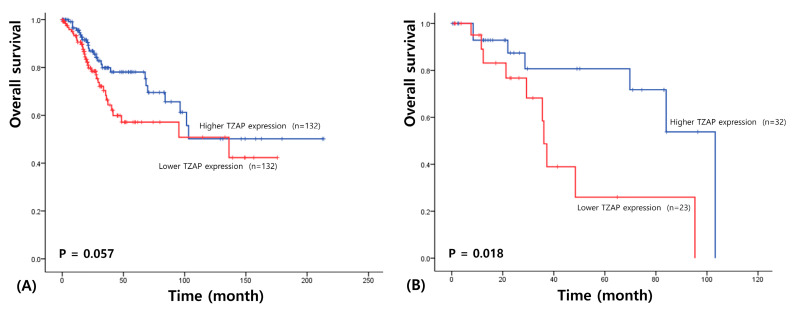
Survival analysis of cervical cancer. (**A**) Overall survival based on TZAP expression in cervical cancer. (**B**) Overall survival based on TZAP expression in N1 stage cervical cancer.

**Table 1 medicina-56-00207-t001:** Clinical characteristics of TZAP expression in cervical cancer.

	TZAP Expression	
	High (%, N)	Low (%, N)	*p* Value
Age			0.275
<65 years	48.4 (103)	51.6 (110)	
≥65 years	56.9 (29)	43.1 (22)	
T stage			0.251
T1	55.4 (72)	44.6 (58)	
T2	50.8 (32)	49.2 (31)	
T3	42.9 (6)	57.1 (8)	
T4	16.7 (1)	83.3 (5)	
N stage			0.337
N0	50.4 (62)	49.6 (61)	
N1	58.2 (32)	41.8 (23)	
M stage			0.641
M0	52.0 (53)	48.0 (49)	
M1	42.9 (3)	57.1 (4)	
Lymphovascular invasion			0.428
(+)	53.2 (42)	46.8 (37)	
(-)	59.7 (40)	40.3 (27)	
Human papilloma virus			0.513
(+)	56.5 (13)	43.5 (10)	
(-)	49.4 (119)	50.6 (122)

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
