# Peer review of "Clinical and Prognostic Significance of TZAP Expression in Cervical Cancer"

_medicina, 2020, doi:10.3390/medicina56050207_

Round 1

Reviewer 1 Report

Dear Authors, 

The paper is overall well written in a comprehensible english.

The paper subject is original as it has not been ever extensively studied.

However, the results of your analysis are really embrionic as they are performed on an already published dataset and they have not been tested on real practice patients.

Moreover, the staging classifications that was used ( the N status) is not used at the present time in the clinical practice.

In my opinion, the paper cannot be published in its present form as it is incomplete in many parts and its results are not valuable for the daily clinical or research practice.

Due to the interesting and unexplored subject it may accepted in a future time, after an analysis on real life patients is performed.

Author Response

The paper is overall well written in a comprehensible english.

The paper subject is original as it has not been ever extensively studied.

--> Thank you for your kind response.

However, the results of your analysis are really embrionic as they are performed on an already published dataset and they have not been tested on real practice patients.

Moreover, the staging classifications that was used ( the N status) is not used at the present time in the clinical practice.

-->  I agree with your opinion about little clinical value of N stage. It was added in Discussion.  

In my opinion, the paper cannot be published in its present form as it is incomplete in many parts and its results are not valuable for the daily clinical or research practice.

Due to the interesting and unexplored subject it may accepted in a future time, after an analysis on real life patients is performed.

--> This is preliminary study and it is the limitation of this article. This description was added.

Reviewer 2 Report

This study to examine TZAP expression in cervical cancer.

TZAP expression may be important for telomere in cancer cells.

TZAP may have an important role in determining cervical cancer prognosis.

Authors focused on cevical cancer to identify the clinical and prognostic value of TZAP expression. Contrary to their expectations, they found that TZAP expression did not have any clinico-pathological or prognostic significance. However, only in N1 stage patients a lower TZAP expression was associated with a poorer survival, which is consistent with the hipothesis that an insufficient TZAP expression could lead to the generation of long telomeres, thus promoting cancer development.

Author Response

This study to examine TZAP expression in cervical cancer.

TZAP expression may be important for telomere in cancer cells.

TZAP may have an important role in determining cervical cancer prognosis.

Authors focused on cevical cancer to identify the clinical and prognostic value of TZAP expression. Contrary to their expectations, they found that TZAP expression did not have any clinico-pathological or prognostic significance. However, only in N1 stage patients a lower TZAP expression was associated with a poorer survival, which is consistent with the hipothesis that an insufficient TZAP expression could lead to the generation of long telomeres, thus promoting cancer development.

--> Thank you for you kind response. An insufficient TZAP expression may lead poor prognosis via over long telomere.

Reviewer 3 Report

In this paper, the authors looked at how expression levels of TZAP is correlated with different prognosis in cervical cancer patients. The idea is novel and interesting as TZAP is a protein involved in telomere maintenance. But overall there are still a few critical points that the authors need to address before the paper could be published.

The introduction needs more information on TZAP. What is the function of this protein? Why overexpression lead to shortened telomeres? Any mechanisms revealed by previous studies? What is the link between TZAP and cancer? Why specifically look at TZAP expression in cervical cancer but not in other cancers? Is TZAP expression higher in cervical cancer than other types of cancers? To address this, the authors should do a comprehensive analysis of TZAP expression levels in a number of different cancer types, and choose to look at those that show higher expression levels, differential expression levels compared to normal tissues, or related to poor prognosis, rather than just “pick” cervical cancer to look at. In fact the study started with looking at TZAP expression levels in different cancer types, but this was not clearly mentioned in the introduction. The introduction should be rewritten to provide the rationale, design, and the most important outcome and impact of the study.

The first part of the paper looked at prognostic values of TZAP across different cancer types. Rather than going into the analysis of prognostic values of TZAP, the authors should first compare TZAP expression levels in normal and tumor tissues in these different cancer types. In addition, rather than presenting the score, the authors should show the Kaplan-Meier curve for the prognosis analysis comparing TZAP high and TZAP low cohort in terms of overall survival, progression free survival etc. Only showing the rank and p value does not seem meaningful as a predictor for prognosis, especially the authors did not even explain what does the rank mean. Without comprehensive analysis, the reason for studying TZAP in cervical cancer cannot be justified.

For the analysis of TZAP in cervical cancer patients, the authors did not show relevant information of how they stratify the patients for the analysis, how many patients in each group, their age, ethnicity, initial diagnosis ect.

Rather than looking at TZAP along, the authors should also look at other gene expressions and parameters related to telomere maintenance and the functional consequences of TZAP expression levels. Such as other telomere associated genes that could also affect telomere length, genetic signatures related to different mechanisms of telomere maintenance, as well as telomere length itself if data is available. If different cervical tumors use different mechanisms to maintain telomere length, this would affect the final outcome of the data analysis if samples were not grouped correctly. As far as I know, cervical cancers usually do not have Tert promoter mutations. But do they have higher telomerase expression level due to amplifications or other mechanisms that activate telomerase transcription? Or do cervical cancers activate the ALT pathway for telomere maintenance?

Author Response

In this paper, the authors looked at how expression levels of TZAP is correlated with different prognosis in cervical cancer patients. The idea is novel and interesting as TZAP is a protein involved in telomere maintenance. But overall there are still a few critical points that the authors need to address before the paper could be published.

The introduction needs more information on TZAP. What is the function of this protein? Why overexpression lead to shortened telomeres? Any mechanisms revealed by previous studies? What is the link between TZAP and cancer? Why specifically look at TZAP expression in cervical cancer but not in other cancers? Is TZAP expression higher in cervical cancer than other types of cancers? To address this, the authors should do a comprehensive analysis of TZAP expression levels in a number of different cancer types, and choose to look at those that show higher expression levels, differential expression levels compared to normal tissues, or related to poor prognosis, rather than just “pick” cervical cancer to look at. In fact the study started with looking at TZAP expression levels in different cancer types, but this was not clearly mentioned in the introduction. The introduction should be rewritten to provide the rationale, design, and the most important outcome and impact of the study.

--> The detail description about TZAP was added in Introduction. And the rationale and design of the study was added.

The first part of the paper looked at prognostic values of TZAP across different cancer types. Rather than going into the analysis of prognostic values of TZAP, the authors should first compare TZAP expression levels in normal and tumor tissues in these different cancer types. In addition, rather than presenting the score, the authors should show the Kaplan-Meier curve for the prognosis analysis comparing TZAP high and TZAP low cohort in terms of overall survival, progression free survival etc. Only showing the rank and p value does not seem meaningful as a predictor for prognosis, especially the authors did not even explain what does the rank mean. Without comprehensive analysis, the reason for studying TZAP in cervical cancer cannot be justified.

--> The effect of TZAP was different in cancers. Therefore, we selected cancer type by RANK (Its meaning was introduced in Method part). Then, we compare TZAP expression levels in normal and tumor tissues in CC. The stream of this study was revised in Introduction

For the analysis of TZAP in cervical cancer patients, the authors did not show relevant information of how they stratify the patients for the analysis, how many patients in each group, their age, ethnicity, initial diagnosis ect.

--> CC patients were divided according to the median value of TZAP expression (high vs. low expression). It was added in Method part. And possible information about CC patients was provided. In Table 2.

Rather than looking at TZAP along, the authors should also look at other gene expressions and parameters related to telomere maintenance and the functional consequences of TZAP expression levels. Such as other telomere associated genes that could also affect telomere length, genetic signatures related to different mechanisms of telomere maintenance, as well as telomere length itself if data is available. If different cervical tumors use different mechanisms to maintain telomere length, this would affect the final outcome of the data analysis if samples were not grouped correctly. As far as I know, cervical cancers usually do not have Tert promoter mutations. But do they have higher telomerase expression level due to amplifications or other mechanisms that activate telomerase transcription? Or do cervical cancers activate the ALT pathway for telomere maintenance?

--> Telomere may be regulated by TERT or ALT pathway. Our data provided the clinical value of TZAP in cancers, especially CC only. To clarify this mechanism, cell line study should be performed. This limitation was added in Discussion

Reviewer 4 Report

Dear Editor, 

in this work the Authors present data in the same line of their previous work published in 2019. 

Their data is new, comprehensible and should bring interest in their field. I only have small recommendation on adding some data that i found missing. This data (mostly precision and a comparison with other telomeric factor) would help the readers to put the authors findings into a more precise context. 

all the best

Author Response

in this work the Authors present data in the same line of their previous work published in 2019. 

Their data is new, comprehensible and should bring interest in their field. I only have small recommendation on adding some data that i found missing. This data (mostly precision and a comparison with other telomeric factor) would help the readers to put the authors findings into a more precise context. 

--> I agree that TZAP may play with other telomere factors closely. This description was added in Discussion.

Round 2

Reviewer 1 Report

The paper in the present form is better.

We are waiting for results in real patients

Author Response

The paper in the present form is better.

We are waiting for results in real patients

--> Thank you for your kind response. We are prepared IRB and cervical cancer samples for this study. It will be performed further.  

Reviewer 3 Report

In the revised manuscript, the authors addressed some of the major problems of the paper, but there are still a few points that needs to be improved.

In the introduction section, I would suggest to provide more background information of TZAP and the rationale and leading hypothesis of this study. The revised manuscript did improved compared to the original submission, but only one sentence and one reference about TZAP function was added, and no hypothesis was put forward based on the previous studies about TZAP, which might be confusing to readers.

In results section 3.1, the authors described: "TZAP 68 expression showed a good prognostic value in bladder, breast, colon, and pancreatic cancers. While 69 it did not show a high statistical value for CC genes, its P-value was particularly high for this cancer 70 type (No. 158 of all registered cancers)." Here I thought the authors should mean rank, rather than P value. And it did not state what does the high and low rank/p value mean.

Author Response

In the revised manuscript, the authors addressed some of the major problems of the paper, but there are still a few points that needs to be improved.

In the introduction section, I would suggest to provide more background information of TZAP and the rationale and leading hypothesis of this study. The revised manuscript did improved compared to the original submission, but only one sentence and one reference about TZAP function was added, and no hypothesis was put forward based on the previous studies about TZAP, which might be confusing to readers.

--> The hypothesis was added in Introduction.

In results section 3.1, the authors described: "TZAP 68 expression showed a good prognostic value in bladder, breast, colon, and pancreatic cancers. While 69 it did not show a high statistical value for CC genes, its P-value was particularly high for this cancer 70 type (No. 158 of all registered cancers)." Here I thought the authors should mean rank, rather than P value. And it did not state what does the high and low rank/p value mean.

-->Our description was weak. Addiationally, the defination of P value and Rank was descripbed in Methods.

Reviewer 4 Report

This work stems from findings made in 2017 by Li et al., Interestingly, the authors performed investigations discussed in previous works: first, comments made by Donati et al., 2017., a brief report by the authors on Heo et al., 2018 and an earlier study on Breast cancer by Heo et al., 2019.

Regarding the methods and results, I believe the authors need to precise the following points:

They compared 21 cancer types according to the TZAP expression. This data is presented below in a table format. Can the author present their data using a more visual design (e.g., A graph could be nice).

Additionally, regarding this data and the cancer types presented, I wonder if the Gender bias issue was examined (e.g, sex ratio) can the author develop on this point, as this might introduce some bias, one way or another.

Data from table 2, could be discussed a little further, at it seems that the High TZAP expression % seem to decrease with stages (with the limitation of the T4 having only one individual). Again, here a graph to appreciate the repartition of the expression could be a valuable information.

The authors are really brief on the survival analysis; I wonder if they can also present the survival analysis using other telomeric factors (this will help with context). Especially knowing that TZAP competes with TRF1 and TRF2. Moreover, data on telomere length should be acknowledge somewhere. One could hypothesizes that the association between telomere length and TZAP expression can provide a good indicator regarding CC.

Last, I have several comments regarding the writing of this piece:

Lines 37-38: Correct me if I’m wrong but TZAP is mostly defined now as a telomere trimming protein, not shortening (e.g., it trims to a define telomere length, but not beyond). Hence, here the term shortening could be misleading.

Line 40: Authors report the lack of study regarding TZAP and cancer, I am not sure to understand why the authors here a disregarding their own previous works: Heo et al., 2017 and 2019 (in Medicina by the way).

Line 98-99: This sentence is misleading, all cancers do not present long telomeres (i.e., see Barthel et al., 2017; Suraweera et al., 2016 among many others). If telomeres are longer in tumor cells than telomeres from their neighboring tissues; they can’t be defined as absolute “long” telomeres without this reference.

Lines 109-114: These few sentences are reporting previous study in agreement with the authors study. While I see no issue there, I think it is not honest to talk simply about “previous” study when it is in fact the authors work. They should clarify this point.  

Author Response

This work stems from findings made in 2017 by Li et al., Interestingly, the authors performed investigations discussed in previous works: first, comments made by Donati et al., 2017., a brief report by the authors on Heo et al., 2018 and an earlier study on Breast cancer by Heo et al., 2019.

Regarding the methods and results, I believe the authors need to precise the following points:

They compared 21 cancer types according to the TZAP expression. This data is presented below in a table format. Can the author present their data using a more visual design (e.g., A graph could be nice).

--> Visual design is better, so we revised it as Figure.

Additionally, regarding this data and the cancer types presented, I wonder if the Gender bias issue was examined (e.g, sex ratio) can the author develop on this point, as this might introduce some bias, one way or another.

-->It is possible because of cervical cancer. This point was added in Discussion.

Data from table 2, could be discussed a little further, at it seems that the High TZAP expression % seem to decrease with stages (with the limitation of the T4 having only one individual). Again, here a graph to appreciate the repartition of the expression could be a valuable information.

--> TZAP seem to decrease with stages, however, it did not have significance because of rarity of T4 stage.  

The authors are really brief on the survival analysis; I wonder if they can also present the survival analysis using other telomeric factors (this will help with context). Especially knowing that TZAP competes with TRF1 and TRF2. Moreover, data on telomere length should be acknowledge somewhere. One could hypothesizes that the association between telomere length and TZAP expression can provide a good indicator regarding CC.

--> I agree about this approach. However, TCGA data have no data about telomere length. Translational research should be performed further. This consideration was described in Discussion.

Last, I have several comments regarding the writing of this piece:

Lines 37-38: Correct me if I’m wrong but TZAP is mostly defined now as a telomere trimming protein, not shortening (e.g., it trims to a define telomere length, but not beyond). Hence, here the term shortening could be misleading.

--> It was revised.

Line 40: Authors report the lack of study regarding TZAP and cancer, I am not sure to understand why the authors here a disregarding their own previous works: Heo et al., 2017 and 2019 (in Medicina by the way).

--> These references were added.

Line 98-99: This sentence is misleading, all cancers do not present long telomeres (i.e., see Barthel et al., 2017; Suraweera et al., 2016 among many others). If telomeres are longer in tumor cells than telomeres from their neighboring tissues; they can’t be defined as absolute “long” telomeres without this reference.

--> It was revised.

Lines 109-114: These few sentences are reporting previous study in agreement with the authors study. While I see no issue there, I think it is not honest to talk simply about “previous” study when it is in fact the authors work. They should clarify this point.  

--> It was revised as our study.